# Modelling Leverage of Different Soil Properties on Selenium Water-Solubility in Soils of Southeast Europe

Lucija Galić [1], Vlatko Galić [2,*], Vladimir Ivezić [1], Vladimir Zebec [1], Jurica Jović [1], Mirha Đikić [3], Adrijana Filipović [4], Maja Manojlović [5], Åsgeir Rossebø Almås [6] and Zdenko Lončarić [1]

1   Faculty of Agrobiotechnical Sciences Osijek, Josip Juraj Strossmayer University of Osijek, Vladimira Preloga 1, 31000 Osijek, Croatia
2   Agricultural Institute Osijek, Južno Predgrađe 17, 31000 Osijek, Croatia
3   Faculty of Agriculture and Food Science, University of Sarajevo, 71000 Sarajevo, Bosnia and Herzegovina
4   Faculty of Agriculture and Food Technology, University of Mostar, 88000 Mostar, Bosnia and Herzegovina
5   Faculty of Agriculture, University of Novi Sad, 21000 Novi Sad, Serbia
6   Faculty of Environmental Sciences and Natural Resource Management, Norwegian University of Life Sciences, Universitetstunet 3, 1430 Ås, Norway
*   Correspondence: vlatko.galic@poljinos.hr

**Abstract:** Selenium (Se) is a nonmetal that is essential for humans and other animals, and is considered beneficial for plants. The bioavailability of Se strongly influences its content in the food chain. Soils are the main source of Se, and their Se content primarily influences its availability, along with other soil properties. A field survey was conducted on soils of Southeast Europe, specifically in Croatia (Osijek), Bosnia and Herzegovina (Sarajevo, Banja Luka, Mostar, and Prud), and Serbia (Novi Sad). Soil samples were taken from the arable soil layer (0–30 cm depth), and two types of Se availability were measured: Se extracted using pure $HNO_3$ ($Se_{Tot}$) and Se readily extracted in water ($Se_{H_2O}$). Only soils from the Mostar area had Se concentrations above deficit levels (0.5 mg kg$^{-1}$), with the highest values of cation exchange capacity (CEC), soil organic matter (SOM) measured as loss of ignition (LOI), total C, total N, $Zn_{Tot}$ and $Cd_{Tot}$. The connections between the chemical characteristics of the soil and $Se_{H_2O}$ were investigated. Principal component analysis (PCA) explained 73.7% of the variance in the data set in the first three principal components (PCs). Using the provided data, we developed a partial least squares (PLS) regression model that predicted the amount of $Se_{H_2O}$ in the soil, with an accuracy ranging from 77% to 90%, depending on the input data. The highest loadings in the model were observed for LOI, CEC, total C, total N, and $Se_{Tot}$. Our results indicate the need for biofortification in these key agricultural areas to supplement the essential dietary requirements of humans and livestock. To efficiently and economically implement biofortification measures, we recommend utilizing regression models to accurately predict the availability of Se.

**Keywords:** selenium; soil properties; deficiency; regression model/analysis; Southeast Europe; biofortification

## 1. Introduction

Selenium (Se) is an essential nonmetallic trace element for humans and animals [1] with a narrow range between deficiency and toxicity (40–400 µg per day). Selenium has numerous functions in the human body such as in the antioxidant defense system and oxidative metabolism, thyroid hormone metabolism, the immune system, male fertility, the prevention of cancer and cardiovascular diseases [2,3]. Furthermore, it was reported that in humans, Se stimulates the uptake of Fe in combination with Mn, Zn, and Cr [4]. In general, higher animals and humans directly acquire organic Se compounds, but they can also internalize inorganic Se [5]. Se deficiency has even been linked with Parkinson's disease mortality in the present day [6]. Selenium deficiency in soils often results in low Se concentrations in foods and negatively affects human and animal health [7]. Globally, Se deficiency is a more common problem from a human health point of view than toxicity [8].

Deficiencies in Se occur in numerous countries across Europe, Asia, South America, and Africa. Specifically, countries including Austria, Belgium, Croatia, Czech Republic, Denmark, Finland, France, Germany, Greece, Ireland, Italy, Poland, Portugal, Serbia, Slovakia, Slovenia, Spain, Turkey, and the United Kingdom in Europe; China, India, Nepal, and Saudi Arabia in Asia; Brazil, Argentina, and Uruguay in South America; Burundi and Egypt in Africa; and Papua New Guinea in Oceania, have reported instances of Se deficiency [9]. Agricultural soils frequently experience selenium deficiency, which is expected to worsen due to changes in weather patterns, including longer periods of drought and more heatwave days [10]. Although not yet proven essential as a plant nutrient, Se has a positive impact on plant growth and quality by aiding in plant antioxidant pathways. As a crucial component of human and animal diets, selenium is primarily sourced from the soil and transferred to the food chain through plants [11]. Se intake varies mostly depending on the Se content in the soil [12] and other site-specific characteristics [13]. Lui et al. (2021) utilized a predictive regression model spanning 55,500 km$^2$ to demonstrate that the distribution of soil Se content is influenced by a variety of factors, including topography, parent material, and climate, which in turn contribute to the development of distinct soil physicochemical properties [14]. As detailed below, the bioavailability of soil Se is related to soil Se content, crop species, and genotype, and soil physicochemical properties such as soil texture, redox potential, pH, and SOM status [15]. Understanding Se bioavailability in soils and its dependence on environmental factors is thus of crucial importance to prevent Se deficiency and improve Se availability in human diets [16]. Hence, experts suggest utilizing the current understanding of selenium behavior in soil to develop predictive models and maps, which can aid in the identification of regions with low selenium availability in soil [10]. Accordingly, the special biological function of Se as a trace element has received attention from researchers in the fields of geochemistry and environment, health, and agricultural sciences [15]. However, limited attention has been paid to the relationship between soils and human health by soil scientists and medical professionals [17]. Therefore, the term geomedicine has been coined to designate the "influence of ordinary natural processes on the health of humans and animals" [17].

Selenium is present in four different oxidation states in aqueous and subsurface system, namely II, 0, IV, and VI, and can easily form compounds with metals, corroborating the fact that it occurs in about 50 minerals [18]. Selenium chemically resembles sulfur and in the geosphere, and it is associated with sulfur deposits and coal [16]. Selenate and sulfate apparently share the uptake system in plants as is evident from their mutual competition [19]. The bioavailability of Se is regulated by the physicochemical conditions of the soils such as the pH, redox conditions, salinity, SOM, etc. [20]. Se in soils occurs in inorganic forms as selenate ($SeO_4^{2-}$) and selenite ($SeO_3^{2-}$), as well as in organic forms [21], such as dimethylselenide (DMSe) and dimethyldiseledide (DMDSe), which can volatize from soils [22]. Selenate, which is water-soluble, represents the most available species of Se for plants in well-aerated, neutral to alkaline soils [23], while selenite is almost entirely unavailable due to its strong adsorption onto soil particles under relevant environmental pH values in nearly all soil types [16,24]. Being a substantial component of soils and sediments, soil organic matter plays an important role in the Se speciation and mobility [16]. Selenium content in soil is influenced mainly by the parent material and climate, with arid and semiarid areas containing larger amount of Se, whereas in humid and irrigated areas, soils show a lower Se content due to leaching [25]. Alkaline soils have more available Se, where it is mostly present in the form of selenate. On the other hand, in acidic, poorly aerated soils, Se occurs mainly as insoluble selenides and elemental Se bound to Fe oxides [26]. Typical Se concentrations of 0.13, 0.05, and 0.5 mg kg$^{-1}$ are reported for ultramafic rocks, mafic rocks and granites, and shales/clays, respectively [16]. Selenium binds to organic and clay soil fractions and is found in phosphates, uranium ore, fossil coal, oil, and shale with a high organic carbon content [26]. Soluble Se is the main source of Se available for plant uptake [27] and its concentrations are usually <0.05 Se µg g$^{-1}$ [26]. SOM can harbor as much as 50% of the total Se in soils, from which a substantial fraction can be mobilized

into soil solution following plant uptake of dissolved Se [28]. SOM plays a crucial role in predicting the availability of Se in soil, as it governs its mobility consistently [29]. The retention of Se in soil is a multifaceted process that is influenced by not only the surface charges of the soil, but also by the presence of anions such as sulfate, nitrate, or phosphate. This is due to the displacement of Se from the soil's adsorption complex [30,31]. As a result, the application of fertilizers can reduce Se retention in the soil as it is substituted by other anions and cations on the soil's adsorption complex [7]. An accurate assessment of Se's fate under field conditions requires information on the rates of Se transformation [32]. Selenium content in soil can vary, but the majority of European soils are poor in Se [33]. The levels of Se in most soils from the Balkans region are low, with concentrations between 0.024 and 0.45 $\mu$g Se g$^{-1}$ [26]. Normal soil Se levels range from 0.1 to 2.0 $\mu$g g$^{-1}$, while toxicity is exerted between 30 and 324 $\mu$g g$^{-1}$ [34], and healthy soils contain around 2 $\mu$g g$^{-1}$ [34]. Generally, total Se levels below 0.5 $\mu$g g$^{-1}$ are considered as deficient [26]. Because dietary intake is the most practical pathway providing sufficient human Se supplies, the biofortification of Se in agriculture through Se fertilization, breeding, or the genetic manipulation of crops has been proposed as an effective and safe measure [35]. Previous research and meta-analyses have determined that biofortification is an effective method for increasing the Se concentration in the most widely cultivated cereals, including wheat, barley, corn, and rice [23]. Biofortification practices have therefore gained increasing attention worldwide in the science-based development of selenate-enriched agricultural products [35,36]. Hence, it is essential to understand soil properties and mechanisms affecting Se uptake by plants to enable accurate predictions of Se status in soils, and develop effective management practices and fertilization recommendations to avoid Se deficiency, Se toxicity, and potential negative environmental impacts [27]. Studies have shown that several soil properties, such as pH, total organic carbon (TOC), CaO, Mn, Mo, and S, significantly affect the uptake and bioaccumulation of heavy metals and Se in rice. Thus, soil properties are generally chosen as independent variables in predictive models [37]. Furthermore, there has been a recent proposal to generate maps using predictive models to identify regions that suffer from Se deficiency in their soils [10].

Biofortification with Se may be an important strategy to provide sufficient Se status in crops [38]. This study was designed to investigate the geochemical factors controlling Se solubility in the water extracts of soils collected from highly productive agricultural regions of Croatia, Bosnia and Herzegovina, and Serbia. We hypothesized that the solubility of Se is mostly controlled by the SOM, CEC, and the pH reaction.

## 2. Materials and Methods

### 2.1. Study Area

This work represents a geochemical study of soils sampled from agricultural fields in Croatia (Osijek), Serbia (Novi Sad), and Bosnia and Herzegovina (Sarajevo, Mostar, Banja Luka and Prud), all located in Southeast Europe. The dominating soil types in Osijek are Chernozems, Eutric Cambisol, Luvisols, and Gleysols [39], and in Banja Luka and Prud are Stagnic Podzoluvisols, Fluvisols, Umbric Gleysols and Eutric Gleysols [40]. In Sarajevo, the predominant soils types are Chromic Luvisols, Eutric Cambisol, Leptosols $\times$ Rendzic Leptosols, and Vertisols, and in Mostar, they are Lithic Leptosols, Regosols, Leptosols $-$ Rendzic Leptosols, Chromic Cambisols, Fluvisols in the river valleys, but Umbric and Eutric Gleysols in the karst fields [40]. The whole part of Novi Sad area lies on a fluvial terrace with the autochthonous soil type, Fluvisol [41].

## 2.2. Sampling and Pretreatment of Soil Samples

In 2015, we conducted field sampling to collect soil samples from the arable soil layer, which was 0–30 cm deep. A diagonal sampling method was employed to collect 20 to 25 punctures on homogeneous plots using a soil probe. We selected 52 sampling sites, comprising 10 from Sarajevo, 5 from Banja Luka, 9 from Novi Sad, 10 from Mostar, 13 from Osijek, and 5 from Prud. These sites were chosen to represent the primary granaries of the region. Samples were dried and sieved through a 2 mm mesh for the determination of soil pH and the trace metal water extraction of Fe, Ni, Cr, Cd and Zn. For the determination of SOM using LOI and total metal extraction using ultra-pure $HNO_3$, samples were further ground to a finer particle size using agate mortar.

## 2.3. Chemical Analysis

The SOM content was estimated by calculating the LOI. The process involved placing oven-dried soil samples in a crucible, which was then ignited in a muffled oven at a temperature of $550 +/− 25 °C$ for a minimum of 3 h. After ignition, the crucible and sample were cooled for 30 min in a desiccator before being weighed. The SOM values obtained were then corrected for clay content [42]. For the determination of dissolved organic carbon (DOC) in soil, air-dry soil samples were weighted into 50 mL centrifuge tubes and 40 mL ultra-pure water (UP-$H_2O$, MilliQ $H_2O$, electric conductivity < 18.2 MS $cm^{-1}$) was added. The tubes were shaken on a linear shaker for two days and centrifuged at 3000 rpm for 30 min. The suspension was passed through 0.45 μm polyethersulfone membrane filters to poly propylene (PP) test tubes. The C concentrations were then determined using a Shimadzu TOC-5000 analyzer (Shimadzu Scientific Instruments, Columbia, MD, USA). The determination of C using the dry C combustion method was based on the thermal decomposition of carbonate minerals in a furnace at a temperature of approximately 1000 °C. In this process, the sample was burned in a purified $O_2$ gas stream, and other gases produced during the combustion were removed before the $CO_2$ absorption lamp reached the sample [43]. The nitrogen determination was performed by placing a soil sample weighing < 200 mg in a tin cuvette. The sample was then burned in an oven at 950 °C using oxygen gas. After the gases stabilized, they passed through two infrared detectors set up to read $CO_2$ and $H_2O$. The nitrogen was reduced, and the $CO_2$ and $H_2O$ were removed, yielding the amount of $N_2$ in the sample [43–45]. The pH and concentrations of Cr, Ni, Se, Cd, Fe, and Zn were determined in the same water extracts. Soil pH also was determined in a soil to water solution ratio of 1:2.5 [46]. CEC was determined using the barium chloride method where 3 g of soil was added to 40 mL of 0.1 M $BaCl_2$, making the soil to solution ratio 1:13 [47]. The total heavy metal (HM) and Se concentrations in soil were determined after digesting the soil in concentrated ultra-pure $HNO_3$ (1:15 solid:solution ratio) via stepwise heating up to 250 °C using a Milestone Ultra clave for 1 h and 15 min. The Cr, Ni, Se, and Cd concentrations in the prepared water and acid extracts were determined using a Perkin Elmer Sciex Elan Inductively Coupled Plasma Mass Spectrometer (ICP-MS), and Fe and Zn using a Perkin Elmer Optima 5300 DV Inductively Coupled Plasma Optic Emission Spectrometer (ICP-OES). The certified reference material (CRM) used was the SRM 2709 [48]. Safe and toxic concentrations of trace elements were determined according to the World Health Organization paper, Trace elements in human nutrition and health [49]. All analyses were conducted at the Norwegian University of Life Sciences (NMBU, Aas).

*2.4. Data Analysis*

Data analyses were carried out using R software version 4.0.2 [50]. Principal component analysis (PCA) was used to screen the dataset by finding the latent (synthetic) variables, i.e., principal components (PCs) made from linear combinations of variables from the original dataset. Individual PCs represent linear statistical models with the scores (distance from the PC origin for every data point), the loadings (variable contributions for each PC), and the residuals. Input parameters were: LOI, pH ($H_2O$), total carbon (TC), total nitrogen (TN), DOC, LOI/TC, Mg (from the CEC), $Cr_{H_2O}$, $Fe_{H_2O}$, $Ni_{H_2O}$, $Se_{H_2O}$, $Cd_{H_2O}$, $Zn_{H_2O}$, $Cr_{Tot}$, $Fe_{Tot}$, $Ni_{Tot}$, $Zn_{Tot}$, $Se_{Tot}$, and $Cd_{Tot}$. Variables showing correlations stronger than 0.91 were considered for discarding from the analysis as redundant. All variables were scaled, centered, and log transformed. The components explaining at least 10% of the variation present in the dataset were analyzed. The same variables were also used in a penalized regression model in a partial least squares (PLS) framework, implemented into the R/pls library. Briefly, PLS aims, similarly to PCA, to explain variability in the dataset by making projections to latent variables. However, there is a considerable difference in the PLS approach, aiming to simultaneously explain the variability in predictors as well as in responses. The model was calibrated in the leave-one-out validation procedure, where $n - i$ samples are taken to calculate the model, while the $i$th observation is used to perform the predictions. The process is repeated until there are $n$ predicted values, which are then correlated to original data, and used to calculate the root mean square error of predictions (RMSEP). The number of components used in the model was selected based on the lowest value of RMSEP in the validation procedure. Additionally, the latent variables (components) from a calibrated best-performing model were used in a mixed model as fixed covariates, along with location, treated as a random effect with an assumption of homogenous variance in the R/lme4 package [51].

## 3. Results

*3.1. Physical and Chemical Soil Properties of the Analyzed Soil Samples*

Table 1 shows means and standard deviations of chemical properties of samples from all studied areas. The soil organic matter (SOM/LOI) content was highest in Mostar soils (9.7%), while the content at other sites ranged from 5.2% to 6.8%. The mean value of pH ($H_2O$) for all locations was slightly alkaline, 7.18. The Mostar area had the highest value of total carbon (4.7%) and total nitrogen (0.36%). The highest total carbon (TC) values were accompanied by the highest values of LOI. Dissolved organic carbon (DOC) varied by location, from the lowest value in Prud (156 mg kg$^{-1}$) to the highest observed value in Banja Luka (352 mg kg$^{-1}$). The mean value for the LOI/TC (total carbon) ratio across all locations was 2.66. The CEC varied considerably by location, with the highest values around Mostar (101,607 cmolc kg$^{-1}$), and the minimum values observed in Banja Luka sites (30,336 cmolc kg$^{-1}$). Table 1 also shows the total concentrations in water extraction ($Cr_{H_2O}$, $Fe_{H_2O}$, $Ni_{H_2O}$, $Se_{H_2O}$, $Cd_{H_2O}$, and $Zn_{H_2O}$) as well as in ultra-pure $HNO_3$ ($Cr_{Tot}$, $Fe_{Tot}$, $Ni_{Tot}$, $Zn_{Tot}$, $Se_{Tot}$, $Cd_{Tot}$). Sarajevo, Banja Luka, Mostar, and Prud had $Cr_{Tot}$ concentrations above the concentrations prescribed by the WHO (WHO 1996). All sample sites had a higher concentration of $Ni_{Tot}$ than the recommended safe level, which is 35 mg kg$^{-1}$, except the Osijek area [52]. The $Zn_{Tot}$ concentrations in all studied locations were lower than the recommended (50 mg kg$^{-1}$) values in soil [52]. In the area of all sample sites, it was found that $Cd_{Tot}$ kg$^{-1}$ amounts were below the maximum permissible concentrations (MPC—0.8 mg kg$^{-1}$) [52]. Samples from the Mostar area showed concentrations of $Se_{Tot}$ (0.5 mg kg$^{-1}$) above deficiency levels [26], while $Se_{Tot}$ deficiency was determined at all other sampling locations. The calculated $SeO_4^2$ concentrations ($n$ = 52) varied, along with the $Se_{Tot}$ and $Se_{H_2O}$, being the highest in soils from Mostar and lowest in soils from Sarajevo and Osijek: Mostar > Prud > Banja Luka = Novi Sad > Sarajevo = Osijek.

**Table 1.** The table shows mean values including ± standard deviation (SD) of important soil parameters measured in soils from Sarajevo, Banja Luka, Novi Sad, Mostar, Prud, and Osijek (*n* = 52).

| | Sarajevo | Banja Luka | Novi Sad | Mostar | Osijek | Prud |
|---|---|---|---|---|---|---|
| pH ($H_2O$) | 6.58 ± 0.722 | 6.14 ± 1.215 | 7.65 ± 0.200 | 7.67 ± 0.101 | 7.16 ± 0.922 | 7.68 ± 0.548 |
| DOC (mg kg$^{-1}$) | 260 ± 77.888 | 352 ± 263.001 | 162 ± 28.185 | 225 ± 49.272 | 175 ± 40.541 | 156 ± 41.593 |
| LOI (%) | 6.535 ± 2.062 | 6.822 ± 2.303 | 5.201 ± 0.2 | 9.755 ± 1.847 | 5.352 ± 1.189 | 6.016 ± 0.731 |
| Total Carbon (%) | 2.45 ± 1.326 | 2.55 ± 1.016 | 1.78 ± 0.931 | 4.71 ± 0.898 | 2.29 ± 0.771 | 2.32 ± 0.089 |
| LOI/TC | 2.878 ± 0.469 | 2.781 ± 0.522 | 3.177 ± 0.599 | 2.081 ± 0.253 | 2.572 ± 0.838 | 2.589 ± 0.223 |
| Total Nitrogen (%) | 0.24 ± 0.099 | 0.25 ± 0.116 | 0.16 ± 0.048 | 0.36 ± 0.115 | 0.17 ± 0.04 | 0.21 ± 0.038 |
| Na (cmol($Na^+$) kg$^{-1}$) | 0.082 ± 0.023 | 0.076 ± 0.01 | 0.159 ± 0.114 | 0.196 ± 0.094 | 0.12 ± 0.067 | 0.106 ± 0.032 |
| K (cmol($K^+$) kg$^{-1}$) | 0.599 ± 0.351 | 0.484 ± 0.22 | 0.584 ± 0.132 | 0.668 ± 0.147 | 0.646 ± 0.125 | 0.734 ± 0.08 |
| Ca (cmol(1/2$Ca^{2+}$) kg$^{-1}$) | 31.44 ± 39.708 | 24.02 ± 26.971 | 37.44 ± 22.897 | 98.2 ± 6.629 | 54.138 ± 36.994 | 63.2 ± 23.636 |
| Mg (cmol(1/2$Mg^{2+}$) kg$^{-1}$) | 2.324 ± 1.868 | 1.512 ± 0.646 | 4.189 ± 1.158 | 2.69 ± 0.44 | 4.046 ± 1.847 | 4.12 ± 1.657 |
| CEC (cmol$^+$ kg$^{-1}$) | 35.364 ± 39.23 | 30.336 ± 24.2 | 42.428 ± 22.79 | 101.607 ± 6.87 | 59.045 ± 37.73 | 68.03 ± 21.96 |
| $Cr_{Tot}$ (mg kg$^{-1}$) | 139.0 ± 188.75 | 205.4 ± 118.70 | 76.55 ± 11.74 | 108.89 ± 23.96 | 76.71 ± 6.198 | 272 ± 30.33 |
| $Cr_{H_2O}$ (mg kg$^{-1}$) | 0.0103 ± 0.009 | 0.0204 ± 0.017 | 0.0066 ± 0.002 | 0.0065 ± 0.002 | 0.0105 ± 0.009 | 0.0272 ± 0.004 |
| $Fe_{Tot}$ (g kg$^{-1}$) | 30.0 ± 10.31 | 32.8 ± 3.90 | 32.8 ± 3.57 | 34.8 ± 5.57 | 30.08 ± 2.32 | 41.2 ± 1.095 |
| $Fe_{H_2O}$ (g kg$^{-1}$) | 0.00122 ± 0.0007 | 0.0012 ± 0.0009 | 0.00026 ± 0.0003 | 0.00028 ± 0.00008 | 0.0019 ± 0.002 | 0.00034 ± 0.0004 |
| $Ni_{Tot}$ mg kg$^{-1}$ | 83.9 ± 115.09 | 97.6 ± 58.50 | 37 ± 6.22 | 76.54 ± 23.68 | 34.67 ± 2.71 | 242 ± 35.64 |
| $Ni_{H_2O}$ (mg kg$^{-1}$) | 0.0873 ± 0.131 | 0.0682 ± 0.055 | 0.0335 ± 0.012 | 0.0375 ± 0.014 | 0.0359 ± 0.024 | 0.0944 ± 0.093 |
| $Cd_{Tot}$ mg kg$^{-1}$ | 0.44 ± 0.1134 | 0.316 ± 0.101 | 0.195 ± 0.030 | 0.511 ± 0.067 | 0.2431 ± 0.036 | 0.386 ± 0.051 |
| $Cd_{H_2O}$ (mg kg$^{-1}$) | 0.00048 ± 0.0005 | 0.00093 ± 0.0008 | 0.000083 ± 0.00005 | 0.000172 ± 0.0001 | 0.000254 ± 0.0003 | 0.0001 ± 0.000008 |
| $Zn_{Tot}$ mg kg$^{-1}$ | 97.9 ± 0.03 | 85.2 ± 0.021 | 67.1 ± 0.006 | 120.7 ± 0.033 | 68.1 ± 0.007 | 116 ± 0.005 |
| $Zn_{H_2O}$ (mg kg$^{-1}$) | 0.0557 ± 0.039 | 0.1016 ± 0.089 | 0.0041 ± 0 | 0.0255 ± 0.014 | 0.0496 ± 0.061 | 0.0182 ± 0.022 |
| **$Se_{Tot}$ mg kg$^{-1}$** | **0.243 ± 0.12** | **0.334 ± 0.011** | **0.25 ± 0.08** | **0.643 ± 0.237** | **0.228 ± 0.081** | **0.426 ± 0.035** |
| **$Se_{H_2O}$ mg kg$^{-1}$** | **0.0085 ± 0.002** | **0.0109 ± 0.003** | **0.0103 ± 0.002** | **0.0175 ± 0.004** | **0.0089 ± 0.002** | **0.0132 ± 0.001** |

Concentrations of $Se_{Tot}$ and $Se_{H_2O}$ are shown in Table 1. The % $Se_{H_2O}$ of the $Se_{Tot}$ by location was fairly consistent: Sarajevo, 3.5%; Banja Luka, 3.26%; Novi Sad, 4.12%; Mostar, 2.72%; Osijek, 3.9%; and Prud, 3.1% (mean 3.1%, *n* = 52). $Se_{Tot}$ and $Se_{H_2O}$ were linearly related across all soils, and the highest amounts of both were found around Mostar ($Se_{Tot}$ = 0.643 mg kg$^{-1}$ and $Se_{H_2O}$ = 0.0175 mg kg$^{-1}$). In contrast, the lowest observed levels of $Se_{Tot}$ were in the vicinity of Osijek (0.22808 mg kg$^{-1}$), and low $Se_{H_2O}$ levels were found around Sarajevo (0.00856 mg kg$^{-1}$) and Osijek (0.0089 mg kg$^{-1}$). The concentration of $Se_{Tot}$ for the rest of the locations are as follows, from higher to lower: Prud > Banja Luka > Novi Sad as well as $Se_{H_2O}$.

### 3.2. Correlation and Principal Component Analysis (PCA) of Analyzed Soil Properties

Correlation analysis showed a strong relationship between all metal water extracts and their total soil concentrations (Figure 1). As expected, LOI, TC, and TN were strongly correlated, also showing a large number of significant positive correlations with other soil properties and metal concentrations. Correlations between soil pH and alkaline/earth metals were mostly significantly positive, while with water extracts of heavy metals were mostly negative. Except for a very strong correlation between $Se_{Tot}$ and $Se_{H_2O}$, both $Se_{Tot}$ and $Se_{H_2O}$ were most strongly positively correlated with LOI, TC, and TN, along with medium to strong positive correlations with CEC, Na, Ca, and the total concentrations of all heavy metals.

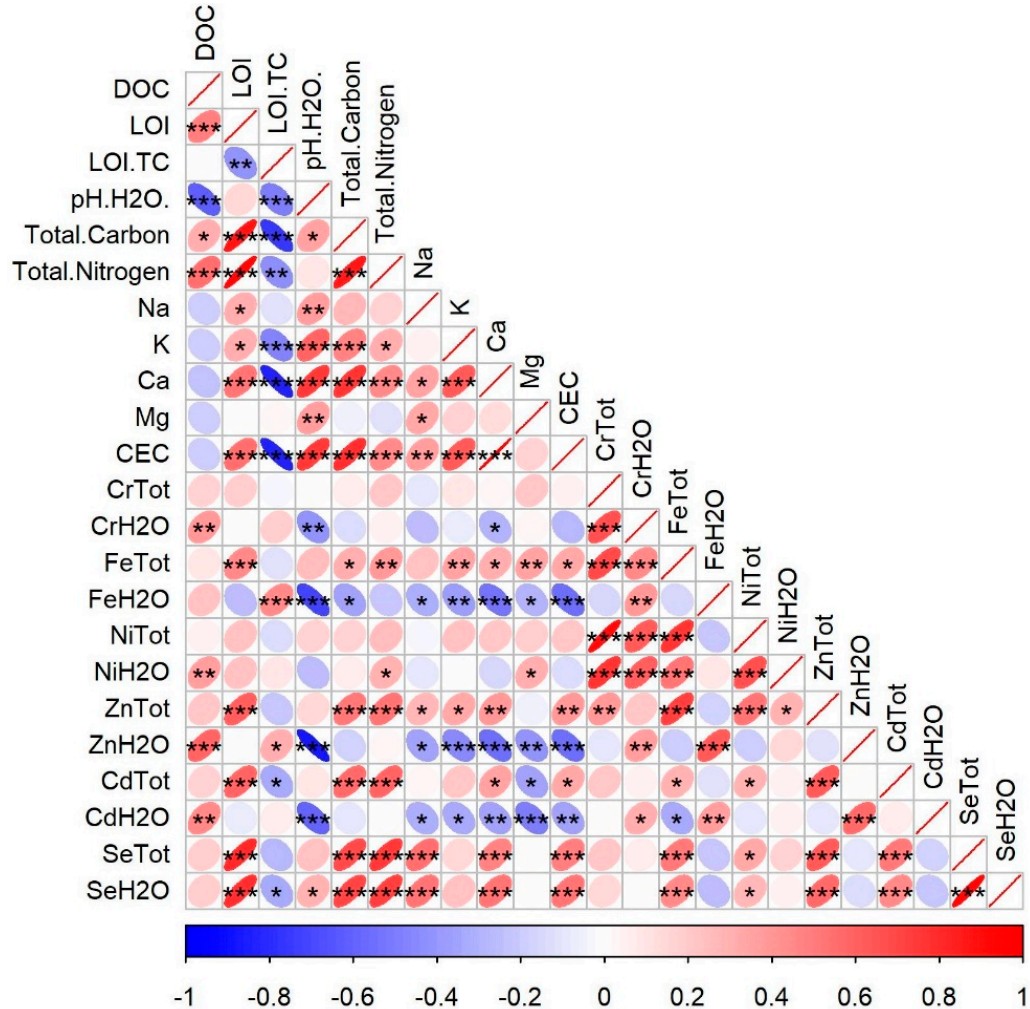

**Figure 1.** The illustration shows the negative (−, blue) and positive (+, red) correlations between soil properties and total soil and total water-extractable concentrations of several elements. Significance of correlation is denoted with * ($\alpha = 0.05$), ** ($\alpha = 0.01$), and *** ($\alpha = 0.001$).

To assess the grouping of soils according to the analyzed properties, PC analysis was carried out. The first three principal components explained 73.6% of the variation in the data set, with 61.1% in the first two components (36.9% by PC1 and 24.2% by PC2, Figure 2). An inspection of the biplot (for communalities see Supplementary Table S1) showed that PC1 was mainly correlated with LOI/TC in the positive direction and TC, CEC, Ca, Mg, Na, and K in the negative direction. PC2 was in a positive correlation with DOC and the water extracts of heavy metals, and in a negative correlation with pH and Mg (Figure 3).

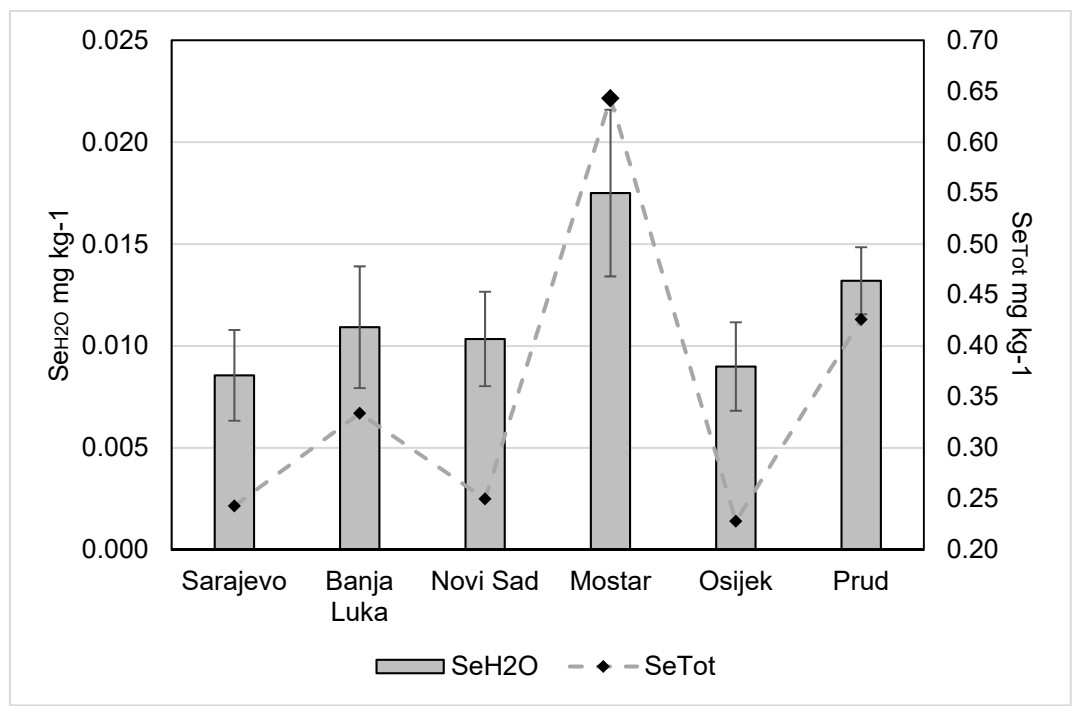

**Figure 2.** $Se_{H_2O}$ concentrations are shown on the left and $Se_{Tot}$ on the right side through studied locations. Location effects were significant in both analyzed parameters at $p < 0.001$ in one-way analysis of variance.

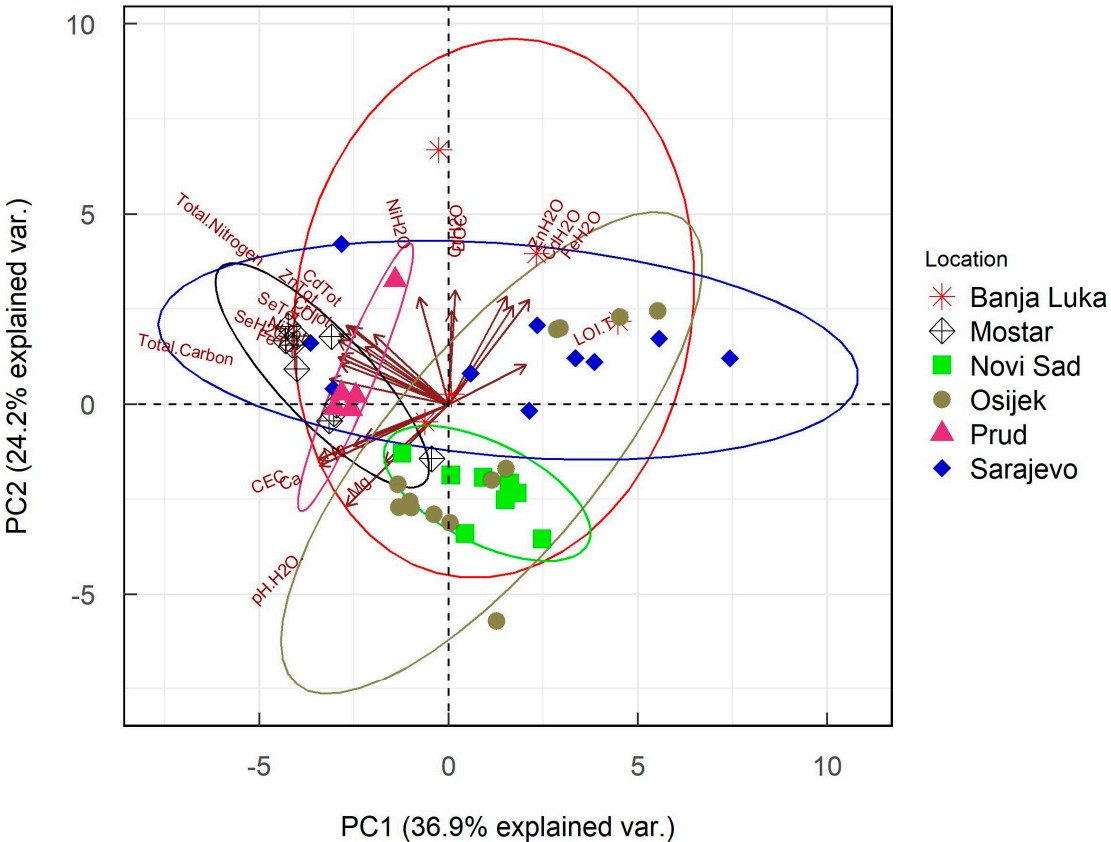

**Figure 3.** Biplot of principal components 1 and 2 from PC analysis of variation among six locations with different soil properties. Arrows represent the eigenvalues of each of the 24 selected soil parameters.

The first and third principal components (PC1 and PC3) explained, together, 49.9% of the variation in the data sets, 12.5% of which was explained by PC3 (Figure 4). An inspection of the biplot (Figure 4) and Supplementary Table S1 shows that PC3 was mainly positive correlated with DOC and negatively with $Cr_{H_2O}$.

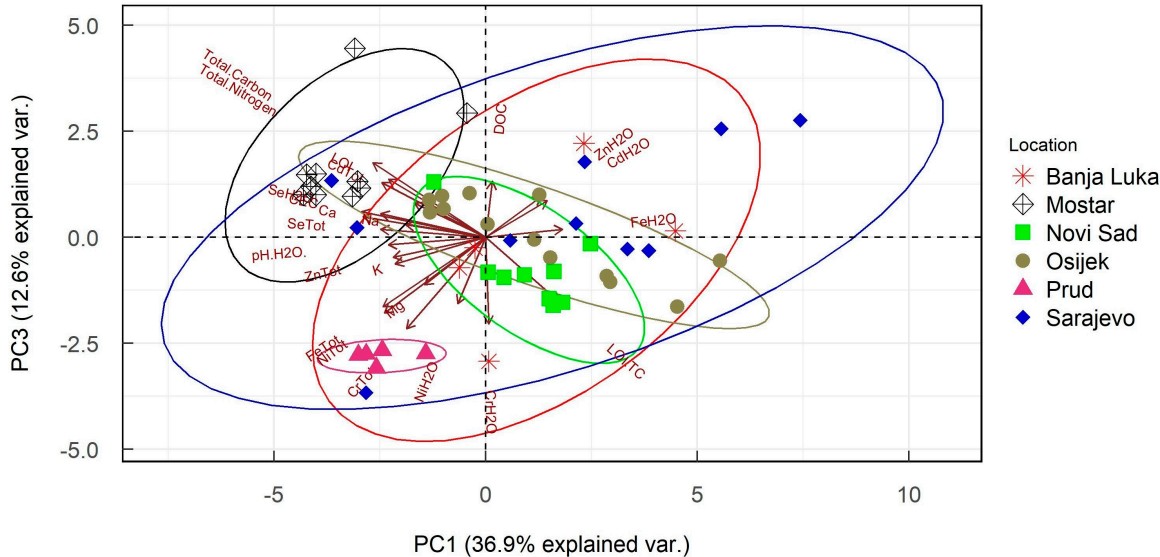

**Figure 4.** Biplot of principal components 1 and 3 from PCA analysis of variation among six locations with different soil properties. Arrows represent the eigenvalues of each of the 24 selected soil parameters.

Generally, soils from Banja Luka, Sarajevo, and Osijek showed considerable variability in their properties, which can be seen from Figures 1 and 2, where ellipses are scattered across all quadrants. In Figure 3, soils from Prud, Novi Sad, and Mostar are more strongly grouped. Samples from Novi Sad also grouped near the origin of PC1, while soils from Prud and Mostar were mostly on the positive side of PC2. Soils from the Osijek sampling location formed two distinct clusters in PC2, with contrasting properties correlated to this component, while the samples from Banja Luka and Sarajevo showed high diversity and appeared scattered across the assessed PCs. Prud, Mostar, and Novi Sad maintained distinctness and tight grouping in PC3 (Figure 4), similar to PC2. Between PC1 and PC3, distinct clusters of samples from Osijek were not visible, while the samples from Banja Luka and Sarajevo showed similar scattering patterns as in PC2. In all three biplots, samples in the top-left quadrant showed the highest $Se_{Tot}$ values. However, it can be seen in Figure 2 that the samples from this quadrant are also accompanied by the higher values of total Ni, Cd, Fe, and Zn. In both biplots, it was shown that only soils from the Mostar region were consistently grouped in quadrants correlated with parameters assessing Se, while soils from other regions showed a grouping influenced by other soil properties.

### 3.3. Regression Analysis of the $Se_{H_2O}$ Concentration

Based on the complex relationships between $Se_{H_2O}$ and the other analyzed soil properties and element concentrations (Figure 1), along with the substantial variability among analyzed soils (Figures 3 and 4), three predictive models for $Se_{H_2O}$ were fitted and calibrated. The first PLS model included all analyzed quantitative properties except $Se_{Tot}$ (Figure 5A), while the second also included $Se_{Tot}$ (Figure 5B). The third model included latent variables from model 2 (Figure 5B) along with a random location effect in a mixed model (Figure 5C). Uncalibrated models 1 and 2 explained 89.78% and 93.82%, respectively, of the variance in $Se_{H_2O}$ (not shown), while the calibrated models explained 77.60%, 86.06%, and 90.20% (Figure 5). As expected, the model without $Se_{Tot}$ (Figure 5A) had a higher error

of prediction, compared to the model with it (Figure 5B). The error of prediction was the lowest in the model 3, including the location random effect (Figure 5C).

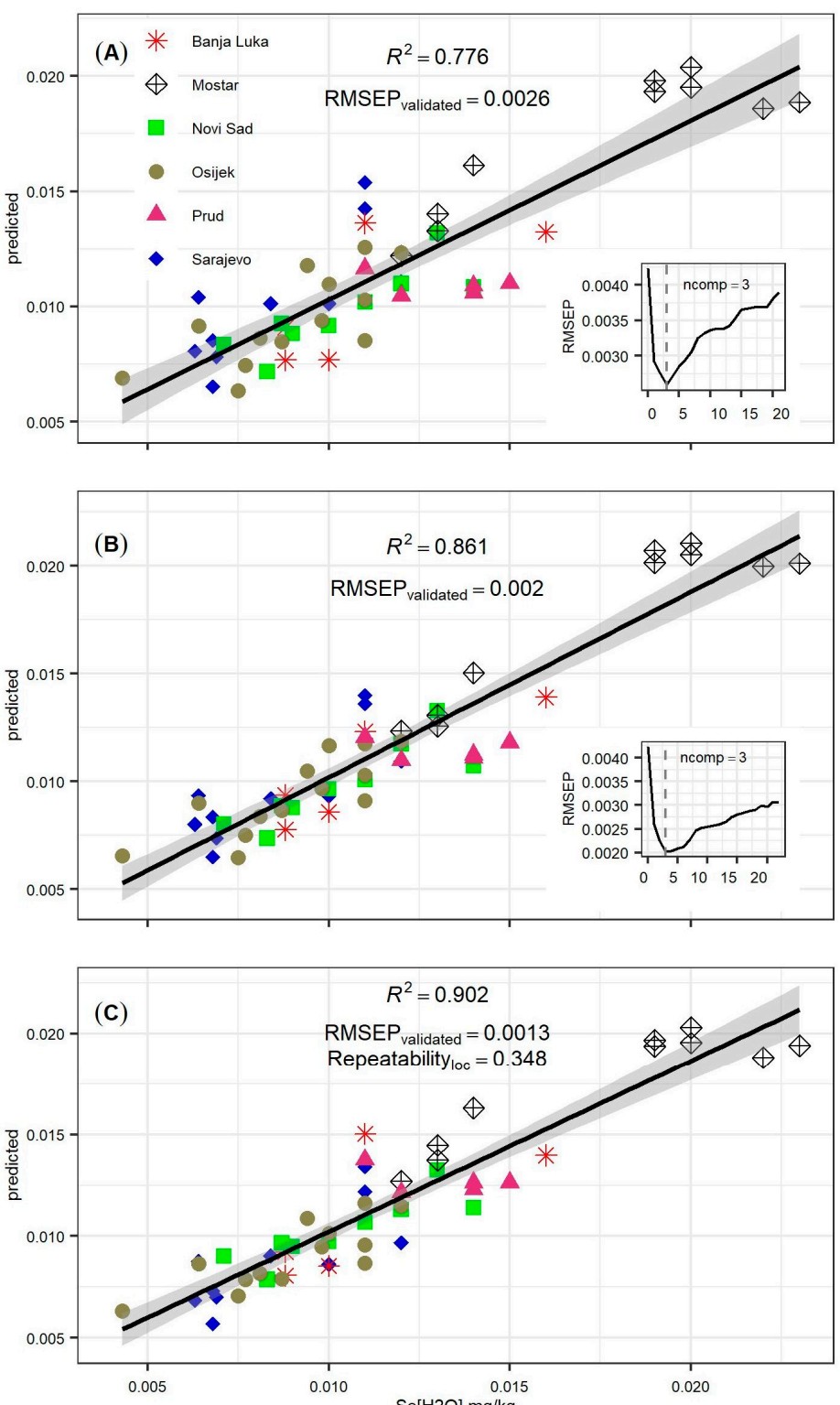

**Figure 5.** Partial least squares (PLS) regression model for $Se_{H_2O}$, including soil properties and concentrations of other elements (**A**), $Se_{Tot}$ (**B**), and PLS scores from (**B**) along with location main effect in a mixed model (**C**). Respective validation errors, the selected number of components, and repeatability of the location random effect are shown within plots.

The loading weights of the first component in models 1 and 2 (Figure 6) mostly resembled the correlations between $Se_{H_2O}$ and other soil properties and concentrations of other elements, with the strongest correlations translated to absolute weights > 0.25 ($Se_{Tot}$, TC, TN, LOI, CEC). The loading weights corresponded to the calculated calibrated model coefficients (Supplementary Figure S1) and variability of a specific variable.

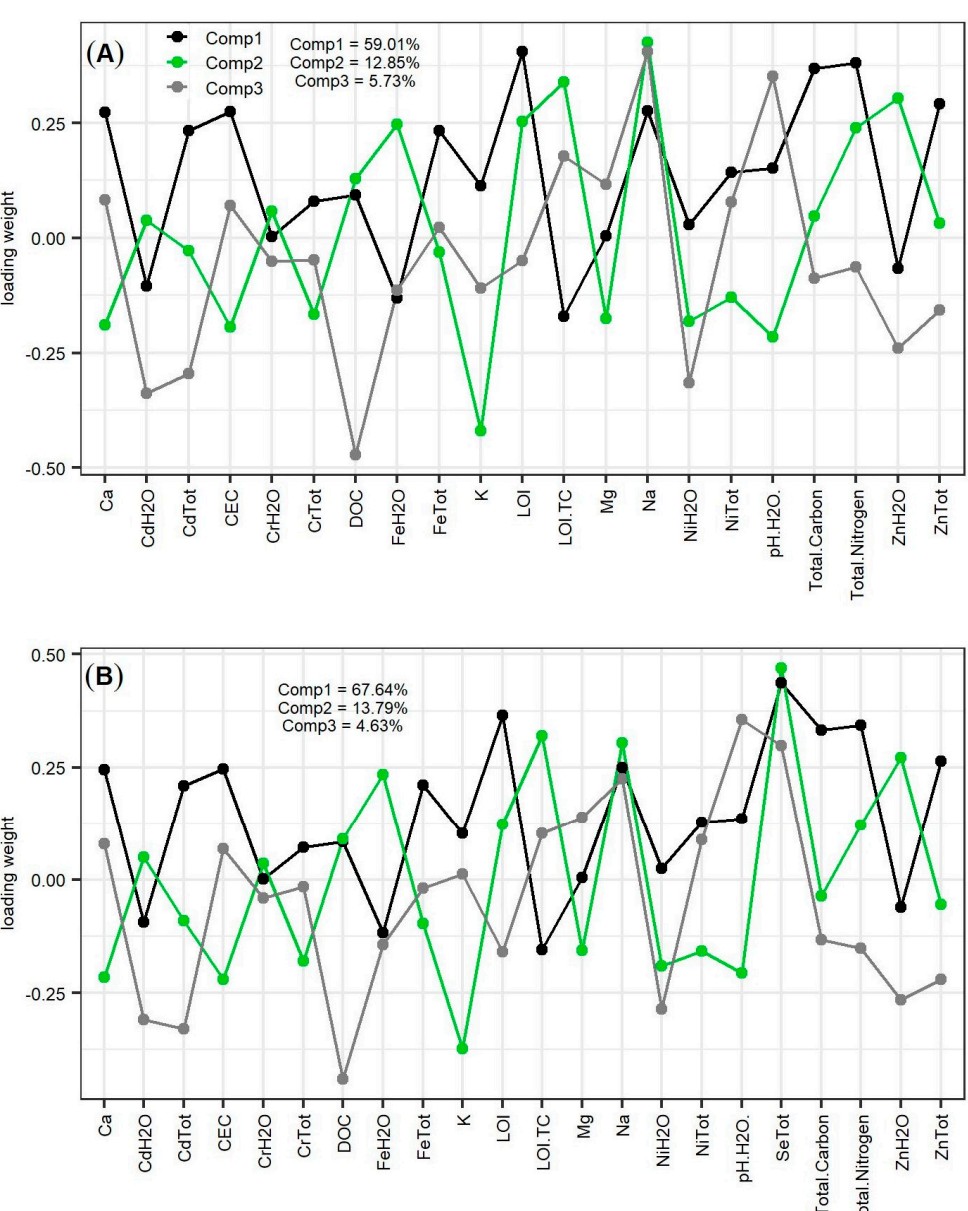

**Figure 6.** Loading weights from the calibrated PLS model (3 components, Figure 5) for $Se_{H_2O}$, including soil properties and concentrations of other elements (**A**) and $Se_{Tot}$ (**B**).

## 4. Discussion

### 4.1. Relationship between Se Concentrations and Soil Physicochemical Properties: A Principal Component Analysis Approach

PC analysis was conducted to assess soil variability at different locations. PCA showed that the soils in the areas of Banja Luka, Sarajevo, and Osijek were the most variable, while on the other hand, soils from the Mostar, Novi Sad, and Prud showed more specific physical and chemical properties (Figures 1–3). Soils from Banja Luka, Sarajevo, and Osijek were not grouped around similar chemical and physical properties. Instead, soils from Sarajevo and Banja Luka showed a scattering pattern across all quadrants in all three biplots (Figures 1–3),

and soils from the Osijek area could be divided to two groups, mostly defined by PC2 (Figures 2 and 3). According to the Soil Atlas of Europe [53], the Osijek agricultural area is mostly represented by Fluvisols and Luvisols in the north and Mollic Gleysols in the south, while the Banja Luka and Sarajevo agricultural regions lie on diverse soils, such as Pellic Vertisols, Stagnic Luvisols, Dystric Cambisols, Chromic Cambisols, and Fluvisols, corroborating the chemical diversity among samples. Soils around Mostar, Novi Sad, and Prud showed more specific characteristics. The Mostar area had the highest values for numerous observed parameters, as well as the Prud area, while the probed sites of Novi Sad area showed the characteristics of poor soils (low LOI, total C, and total N). All sample sites, except Mostar, had the Se levels below the upper deficiency bound (Table 1), with soils containing less than 0.5 $\mu g\ g^{-1}$ of total Se being considered deficient [26]. Physical and chemical properties differed between locations (Table 1). The area of Mostar showed elevated values for the soil properties of LOI, CEC, pH, total C, and total N, and higher concentrations of Ca, Na, $Zn_{Tot}$, $Cd_{Tot}$ and $Se_{Tot}$ and accordingly, $Se_{H_2O}$ (Figures 1 and 2). There is a possibility that the origins of present elements are similar due to the same primary parent rock [54]. A study by Alloway (2013) reported that soils containing Ag, As, Au, Ba, Cd, Cr, Cu, Hg, Mo, Ni, Pb, Sb, Se, Th, Tl, U, V, W, and Zn, and also showing a high content of organic matter and clay, usually originate from black shales, including bituminous and oil shales. This corroborates the findings from our research that Se was more effectively incorporated into soil organic matter at higher values of pH (Figure 1) and is also confirmed in the literature [55]. Several studies have highlighted that the capacity of plants to absorb and store Se is influenced not only by the overall Se concentration in the soil, but also by the bioavailability of Se in the soil [14]. As the pH of soil decreases, the surfaces of clay minerals, organic matter, and metal oxyhydroxides become more positively charged, which is conducive to sorption and the retention of Se oxyanions in soil [3]. Situated along rivers, Mostar, Novi Sad, and Prud share a common feature. While the Neretva River runs alongside Mostar, Novi Sad is located near the Danube River, and the Prud area is situated between the Bosna and Sava Rivers. It is possible that the high levels of pH, Ca, CEC, $Zn_{Tot}$ $Se_{Tot}$ and $Cd_{Tot}$ found in soils from Mostar and Prud are a result of the periodic river floods that affect these areas. This is in accordance with investigation of Pavlović et al. (2016) where it was shown that the riparian soil acts as an important sink for different elements, creating favorable conditions for the seeds of plant species that require a bare soil surface for germination, thus retaining high levels of organic matter, as seen in the Mostar area (the highest LOI). On the other hand, Novi Sad region has a moderately warm humid climate with warm summers [56]. A continental climate is prevalent in Sarajevo [57], Banja Luka [58], and Osijek [59]. Soils in arid and semiarid regions can have a high Se content, whereas those in humid and irrigated regions tend to have lower Se levels [25]. The soil samples from Osijek, Novi Sad, and Sarajevo, which experience cooler and more humid weather conditions in comparison to Mediterranean areas, showed lower levels of Se. The concentration of Se in soils is mainly influenced by climate variables, specifically AI, precipitation, and evapotranspiration, as they play a crucial role in regulating soil leaching processes [10].

*4.2. $Se_{H_2O}$ and $Se_{Tot}$ Concentrations and Correlation between Different Soil Properties and $Se_{H_2O}$*

$Se_{H_2O}$ analysis was performed as it is considered to be a more effective method for assessing Se deficiency than measuring total Se concentration in soil. This is due to the fact that Se concentration in aqueous soil solutions can serve as a more reliable indicator [60]. The water-soluble fraction of soil Se is a function of soil physicochemical properties and varies with biological reactions [60]. Several studies analyzed soil Se availability referring to water-soluble Se, as plant-available, in spite of the effect of exchangeable Se [28]. The proportions of water-soluble Se in the total soil fractions of analyzed soils were very low (Sarajevo, 3.5%; Banja Luka, 3.26%; Novi Sad, 4.12%; Mostar, 2.72%; Osijek, 3.9%; and Prud, 3.1%). The highest concentrations of $Se_{Tot}$ were recorded in soils from the Mostar region, where the water-soluble fraction accounted for the lowest percentage of the total (2.72%).

This may be attributed to the highest content of soil organic matter (SOM), which can bind with Se and decrease its availability. It is known that the availability of Se in soils, similar to sulfur, generally increases with a higher pH value [55]. This trend can be explained by the increased negativity of the surface charges at alkaline pH levels, which causes electrostatic repulsion between the surface and the negatively charged Se oxyanions [61]. Soil organic matter is another important component that retains Se in the soils. The proportion of SOM-bound Se can be affected by the soil type or the specific composition and content of SOM [35]. This was also corroborated by a significant weak to medium positive correlation between soil pH reaction and $Se_{H_2O}$ (Figure 1). It was shown that alkalinity and salinity can induce the precipitation of some elements and also effect adsorption by affecting the CEC [62]. Furthermore, in our study, $Se_{H_2O}$ and CEC showed a significant moderate positive correlation (0.53) (Figure 1). Calcium and Na also showed medium to strong correlations with $Se_{H_2O}$, possibly because they are cations participating in the CEC of alkaline soils. Another perspective given by the recent finding of Xu et al. (2020), indicates that the content of Se and Zn increases in calcareous and ferric soils [63]. The results of Imran et al. (2020) suggest that soils on shale parent rock show the highest CEC and the highest Se content, which is in accordance with our results [54].

Our study revealed a strong positive correlation between $Se_{H_2O}$ and LOI, indicating that the water-soluble fraction of Se increases with a higher content of soil organic matter. This finding is consistent with previous research, which has demonstrated that water-soluble and exchangeable Se tends to increase with advanced weathering, as the argillaceous clay, iron oxides, and organic compounds provide more exchange sites for Se [54]. In our study there was a strong positive correlation detected between $Se_{H_2O}$ and $Zn_{Tot}$ which may be explained by the influence of Se on the bioavailability of Zn in soil. Selenium can affect Zn uptake by plants through processes such as adsorption, oxidation, complexation, and precipitation [1]. Zinc in soil typically occurs in the II oxidation state and its activity is influenced by negatively charged adsorptive surfaces such as SOM, clay, and iron, and manganese hydrous oxides. Furthermore, Zn mobilization can occur through the reductive dissolution of Fe oxides [63]. The weak significant positive correlation detected between $Ni_{Tot}$ and $Se_{H_2O}$ was probably caused by soils with reducing conditions that affect both Se and Ni soil dynamics [64]. Our results show positive moderate to strong correlation between $Fe_{Tot}$ and $Se_{H_2O}$, possibly caused by formation of the organoselenium compounds adsorbed on poorly crystalline iron oxides [54], whereby the OM-bound Se fraction is up to 40–50% [35]. Furthermore, it was found that Fe-oxide-type minerals affect the adsorption of Se [65]. Moreover, amorphous iron is considered to be the most active iron/aluminum oxide and one of the scarce positively charged colloidal minerals present in soil. When in the form of oxygen-containing anions, Se can create a stable inner complex with amorphous iron and co-precipitate with iron hydroxides [14]. Our findings indicate a moderate positive correlation between $Se_{H_2O}$ and $Cd_{Tot}$. This correlation may be attributed to the reduction of Se in the acidic microenvironments of soil or in the rhizosphere, resulting in the formation of selenide that can bind to Cd and form Cd–Se complexes, which can subsequently reduce Cd uptake by root cells [66]. Selenite, selenate, and their products in soil might also thermodynamically react with Cd to form Cd–Se complexes that become unavailable to the plant root [67], marking another health benefit of Se in soil detoxication. Our study showed that agricultural soils in Southeast Europe have low total Se contents, and that Se dynamics are complex and dependent on various soil properties and types. The total Se content in soils seems to be mainly influenced by the soil parent materials and its availability in soil water solutions. As a result, the overall availability of Se appears to be critically low. However, understanding the correlations and factors that affect the distribution and availability of this essential trace element is crucial for advancing the field of geomedicine [17]. Moreover, recent results indicate that the availability of Se might even worsen due to the climate change, especially in Southeast Europe [10].

*4.3. Model for Prediction of Se$_{H_2O}$*

Predicting soil water-extractable Se represents a tentative topic, especially regarding its health benefits in the human diet. In spite of the known high predictive ability of weather patterns for soil Se concentrations [68], our study focused on the predictive ability of the soil geochemical properties with confounding climatological effects. Our study aligns with the findings of Liu et al. (2021), who utilized a predictive model to evaluate Se availability. Their research identified SOM and pH as the key factors impacting model predictability [14]. Our results indicate that the modelling of Se$_{H_2O}$ from soil geochemical parameters in a penalized model could be worthwhile, given the range of soils represented in our study; slightly acidic to slightly alkaline pH, with moderate SOM content (LOI/TC/TN/DOC), with or without information about Se$_{Tot}$ (Figure 5A,B). The addition of Se$_{Tot}$ is expected to increase the model's predictive accuracy due to the established correlation between the extractable and total Se [8], which was also confirmed in our study (Figures 5B and 6B). The study highlighted the significant influence of soil organic matter (SOM) (LOI and TC) on Se extractability in phosphate, which was consistent with our findings in the study of water extracts (shown in Figure 6A,B). Additionally, our study showed that Fe$_{Tot}$ and Ca were crucial in predicting Se concentrations, which was also supported by a study conducted by Xu et al. [63]. This could be caused by the ability of selenite to form inner-sphere complexes, including bidentate and monodentate inner-sphere complexes on the Fe oxide surface [13]. However, Williams Araújo do Nascimento et al. [69] showed that this correlation is valid only in clusters of soils in Fe-rich regions with a specific climate, unlike the stable correlation between Se and SOM across different soils and conditions [70]. Similar findings were also reported by Liu et al. [14], further addressing the importance of aluminum in predicting soil Se content, along with CEC, which was also confirmed in our study. With a comparable sample size to other studies, our study showed that a penalized and mixed model approach might improve predictions of Se$_{H_2O}$, at least in the constrained soils represented by our study. Due to the high importance of Se to human health and its established beneficial effects in plants, more research determining the key parameters for the building of efficient predictive models is needed. The use of predictive models for mapping Se bioavailability and concentration has become increasingly common, as it plays a crucial role in enhancing, progressing, and verifying these maps [37,71]. However, our study represents a promising step in the precise modelling of Se availability, thus facilitating the assessment of the need for measures such as agronomic biofortification.

## 5. Conclusions

Most soils worldwide suffer from Se insufficiency, including all locations examined in this study, except Mostar. This study provides valuable information for Se biofortification in Southeast Europe, where physicochemical properties vary greatly by location. Water-extractable Se, which is available to plants, showed positive correlations with LOI, CEC, total C, total N, Ca, Na, Fe$_{Tot}$ Zn$_{Tot}$ Cd$_{Tot}$ and Se$_{Tot}$. Soils from Mostar, under a Mediterranean climate, had the highest Se content, while soils from other locations with continental, semiarid climates had lower Se contents. The effectiveness of penalized models in predicting water-extractable Se from available geochemical information highlights the dependence of Se concentrations on soil geochemistry, climate conditions, and specific physicochemical properties. Thus, Se biofortification is a necessary step in maintaining a healthy human population in the region. In fact, this predictive model holds great potential for other regions characterized by soil Se deficiency. By making use of basic soil analyses such as SOM, CEC, total N, total C, Ca, and Na, the model's predictive capacity can be significantly enhanced, allowing the acquisition of essential information pertaining to soil Se levels. Ultimately, this approach has the potential to facilitate optimal implementation of Se biofortification, ensuring that an adequate dosage of Se is delivered to crops, and consequently, to people.

**Supplementary Materials:** The following supporting information can be downloaded at: https://www.mdpi.com/article/10.3390/agronomy13030824/s1, Figure S1: Coefficients of calibrated PLS models A and B from Figure 6 in the main text; Table S1: Communalities (correlations between original variables and PCs) from principal component analysis.

**Author Contributions:** Conceptualization, Z.L. and Å.R.A.; methodology, Z.L., Å.R.A. and V.I.; software, V.G.; formal analysis, V.I., V.Z., J.J., M.Đ., A.F., M.M. and L.G.; investigation, L.G., V.G. and Z.L.; resources, Z.L.; data curation, V.G.; writing—original draft preparation, L.G.; writing—review and editing, Z.L. and Å.R.A.; visualization, V.G.; supervision, Z.L. and Å.R.A.; project administration: Z.L.; funding acquisition, Z.L. All authors have read and agreed to the published version of the manuscript.

**Funding:** The present work was financially supported by the project KK.01.1.1.04.0052: "Innovative production of organic fertilizers and substrates (INOPROFS)", co-financed by the European Union from the European Regional Development Fund within the Operational Programme Competitiveness and Cohesion 2014–2020 of the Republic of Croatia. The work of the PhD student Lucija Galić was fully supported by the "Young researchers' career development project–training of doctoral students" through grant HRZZ-DOK-2020-01-1288, financed by the Croatian Science Foundation.

**Data Availability Statement:** All data supporting the conclusions drawn in this manuscript is available from the coauthors upon reasonable request.

**Conflicts of Interest:** The authors declare no conflict of interest.

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
