# Peer review of "Modelling Leverage of Different Soil Properties on Selenium Water-Solubility in Soils of Southeast Europe"

_agronomy, doi:10.3390/agronomy13030824_

Round 1

Reviewer 1 Report

Specific comments:

Abstract

The whole study content did not describe clearly in this part, so it is difficult to catch your point. I think that language needs some revision to make it clearer.

Introduction

The introduction does not cover the existing information satisfactorily. This introduction reads like a popular science article rather than a summary of relevant research. After we read through this introduction, we did not have this feel that it was very necessary to do this work. I think the literature review needs to be more thoroughly and deeply.

Line 43 USis it a redundant word?

Line 45 “increase” should be “increasing”;

Line 57 the citation format of this reference is inconsistent with others;

Line 77-80 this sentence of “in well aerated……….types of soils” should be revised so that it describes clearly;

Material and methods

The whole material and methods were not logically expressed, especially some applications of English tenses and the break of a sentence. Your own present work must be referred to in the past tense.

Results

1.      Line 208 where do you get the coefficient of correlation (0.88)? we can not read it from table 1, you should label its source.

2.      If data listed in your tables (table1) and figures ( figure1) do not make analysis of variance, you can not describe that one is higher than other.  

3.      Where is section 3.2?

4.      Line 239 “shown in figure 4”? or “table 1“

5.      Line 280 Fig. 2?

6.      Line 294 Fig. 2?

Discussion

I find lots of repetitions of the results in this part, and there is a paucity of explanation and support. I think that this part should cite more references to support your results and explain the reasons of these results.

Conclusion

Line 468-471 I do not think these three sentences of common sense knowledge description are appropriate to put in the conclusion. The conclusion should summarize your main findings and discuss the theoretical implications of your research work.

Author Response

Reviewer 1

Dear reviewer, thank you very much for your time and effort in reviewing our manuscript. Below we provide the point-to-point responses to your comments and suggestions. We hope you will find our revisions necessery and of sufficient quality.

Abstract

The whole study content did not describe clearly in this part, so it is difficult to catch your point. I think that language needs some revision to make it clearer.

Thank you for these valuable observations. According to your suggestions, we corrected abstract.

Introduction

The introduction does not cover the existing information satisfactorily. This introduction reads like a popular science article rather than a summary of relevant research. After we read through this introduction, we did not have this feel that it was very necessary to do this work. I think the literature review needs to be more thoroughly and deeply.

Thank you for your suggestion. We hope that the corrected introduction will cover the shortcomings of the previous version of our manuscript.

Line 43 US?is it a redundant word?

Thank you for this observation. We deleted this redundant word.

Line 45 “increase” should be “increasing”;

We have changed whole sentence.

Line 57 the citation format of this reference is inconsistent with others;

Thank you for these valuable observations. The reference now appears correctly in the text of the manuscript.

Line 77-80 this sentence of “in well aerated……….types of soils” should be revised so that it describes clearly;

The sentence has been reconstructed and hopefully more comprehensible.

Material and methods

The whole material and methods were not logically expressed, especially some applications of English tenses and the break of a sentence. Your own present work must be referred to in the past tense.

Thank you for your suggestion. The necessary corrections have been made, and the parts that were previously in the present tense have been transferred to the past tense.

Results

  1. Line 208 where do you get the coefficient of correlation (0.88)? we can not read it from table 1, you should label its source.

We have added the valid information about correlation (0.88). It is in Figure 2.

  1. If data listed in your tables (table1) and figures ( figure1) do not make analysis of variance, you can not describe that one is higher than other.

As the collection of the data did not result from an experimental design, and subsequently, the resulting number of samples is unorthogonal, use of mixed model with random effect of location would be the best fit for analysis of ouur dataset. We tested for the location effect of variance and yielded a non-zero variance component (effects were significant, not shown). However, following Bates (2006, lmer, p-values and all that), the p-values were not appeded, but the standard deviations of each location were given. The aim of reporting the differences was more relying on common sense comparison than retrieving the significance of effects, as locations were scattered over more than 24000 km2.

  1. Where is section 3.2?

This was overseen in the previous version of the paper. We have corrected the error in the revised version.

  1. Line 239 “shown in figure 4”? or “table 1“ ?

Thank you for pointing out the mistake. We apologize for the error and have corrected it in the new version of the paper by adding Table 1.

  1. Line 280 Fig. 2?

The updated version of the paper now includes the corrected version.

  1. Line 294 Fig. 2?

The new version of the paper features the corrected version, resolving the previous error.

Discussion

I find lots of repetitions of the results in this part, and there is a paucity of explanation and support. I think that this part should cite more references to support your results and explain the reasons of these results.

Thenk you for your suggestion, which has helped us improve the paper. We have incorporated your feedback by deleting any repetitions, expanding the discussion section with new references, and providing a more comprehensive explanation of our results.

Conclusion

Line 468-471 I do not think these three sentences of common sense knowledge description are appropriate to put in the conclusion. The conclusion should summarize your main findings and discuss the theoretical implications of your research work.

Thank you for your contribution to the review process of our manuscript. We appreciate your feedback and hope that after our responses you will consider suggesting the manuscript for publication. We acknowledge the need to improve the Conclusion section in Version 1 of our manuscript, and have made the necessary changes based on your suggestions.

Reviewer 2 Report

General comments

It is an interesting manuscript. However, the authors need to improve some things.

In particular, the author should review the abbreviations used in the manuscript. In addition, the authors must make a mineralogical analysis of the soils or associate the availability of selenium to the different particles that are forming the studied soils.

Specific comments

Line 3: The numbers next to the authors should be in superscript.

Line 6: Affiliations must have zip codes.

Line 37: You should improve the introduction. Be careful when using connectors.

Line 52: you should change physio-chemical to physico-chemical.

Line 52: redox potential.

Line 57: The reference (Fairweather-Tait et al., 2011) is not in accordance with the Journal.

Line 68: You must delete + in the oxidation states +IV, and +VI.

Line 72: You can delete the following sentence The concentration of Se in rocks is highly variable depending on rock type [10].

Line 72: You must use the initials SOM on line 52.

Line 91: You must change μg Se g-1 by μg g-1.

Line 92: You can use SOM instead Soil organic matter.

Line 93: you must delete dot after Se.

Line 94-95: I do not understand the sentence; you must improve or deleted it.

Line 106: you can delete selenate or (SeO42-). You should only use one of them.

Line 112: You can delete Thus.

Line 115: You can use SOM and CEC instead of soil organic matter and cation exchange capacity

Line 116-118: You can delete that sentences.

Line 143: You can delete soil organic matter and leave SOM. The same for loss of ignition

Line 210: You must change mg/kg by mg kg-1. You must do this in all your manuscript.

Tale 1: You can delete Ca, Na, K, and Mg and leave only CEC.

Line 330: You must change the subtitle.

Line 358-360: I am not sure about that. Therefore, you should read reference 3 again.

Line 390: You can read about electrostatic repulsion between of Se oxyanion and soil particles.

Line 410: According to those sentences, Se can affect the adsorption of all cations.

Line 468-471: These lines are not part of a conclusion.

Line 468. You must improve your conclusion.

Line 504. You must check in referents used.

Author Response

Reviewer 2

Dear reviewer, thank you very much for your time and effort in reviewing our manuscript. We provide the point-to-point responses to your comments and addressed issues. We hope that after revision, our manuscript will meet the criteria to be published in Agronomy.

Please find our point-to-point responses below:

Line 3: The numbers next to the authors should be in superscript.

Thank you for these observations. According to your suggestions, we corrected numbers next to the authors.

Line 6: Affiliations must have zip codes.

Thank you for this observation. We added zip codes.

Line 37: You should improve the introduction. Be careful when using connectors.

Thank you for these valuable observations. According to your suggestions, we corrected introduction.

Line 52: you should change physio-chemical to physico-chemical.

Thank you for pointing out the mistake. We corrected this mistake.

Line 52: redox potential.              

Thank you for this note. We corrected this in new version of our manuscript.

Line 57: The reference (Fairweather-Tait et al., 2011) is not in accordance with the Journal.

Thank you for this observation. According to your suggestions, we corrected the reference.

Line 68: You must delete + in the oxidation states +IV, and +VI.

We deleted + in the oxidation states.

Line 72: You can delete the following sentence The concentration of Se in rocks is highly variable depending on rock type [10].

Thank you for your suggestion. We deleted this sentence.

Line 72: You must use the initials SOM on line 52.

We have changed this by leaving only SOM in the sentences.

Line 91: You must change μg Se g-1 by μg g-1.

Thank you for this observation. According to your suggestions, we corrected this.

Line 92: You can use SOM instead Soil organic matter.

Thank you for this note. We deleted Soil organic matter and put SOM instead.

Line 93: you must delete dot after Se.

Dot has been deleted.

Line 94-95: I do not understand the sentence; you must improve or deleted it.

Thank you for this valuable suggestion. We have changed the sentence and I hope it will be clearer in the new version of our manuscript.

We deleted SeO42-.

Line 112: You can delete Thus.

We deleted Thus.

Line 115: You can use SOM and CEC instead of soil organic matter and cation exchange capacity

Thank you for this observation. According to your suggestions, we corrected this.

Line 116-118: You can delete that sentences.

Thank you for suggestion. We deleted this sentence.

Line 143: You can delete soil organic matter and leave SOM. The same for loss of ignition

Thank you for your suggestion. We deleted soil organic matter and loss of ignition.

Line 210: You must change mg/kg by mg kg-1. You must do this in all your manuscript.

mg/kg was changed to mg kg-1 throughout the text.

Tale 1: You can delete Ca, Na, K, and Mg and leave only CEC.

We appriciate this suggestion. However, we would like to leave the cation configuration of the soils which is also informative about the parent rock material.

Line 330: You must change the subtitle.

Thank you for these valuable observations. According to your suggestions, we corrected subtitle.

Line 358-360: I am not sure about that. Therefore, you should read reference 3 again.

We are grateful for your suggestion, which has helped us improve this sentence. The sentence has been reconstructed and hopefully more comprehensible.

Line 390: You can read about electrostatic repulsion between of Se oxyanion and soil particles.

Thank you for these valuable observations. According to your suggestions, we corrected sentences and add new reference.

Line 410: According to those sentences, Se can affect the adsorption of all cations.

This was the oversight in the previous version of the paper. It was corrected in the revised version.

Line 468-471: These lines are not part of a conclusion.

Thank you for these observations. According to your suggestions, we corrected this in conslusion section.

Line 468. You must improve your conclusion.

Thenk you for your observetion here, which has helped us improve the manuscript. We have incorporated your feedback by improving conclusion.

Line 504. You must check in referents used.

Thank you for your valuable contribution to the review process of our manuscript. We greatly appreciate your feedback and have carefully considered your suggestions. We are pleased to inform you that we have made the necessary changes to the references in our manuscript based on your insightful feedback. We hope that these revisions meet your criteria and look forward to the possibility of the manuscript being suggested for publication.

Reviewer 3 Report

The article addresses an important topic, is well structured, the data are transformed into important information and the discussion is well grounded.

Some changes indicated in the attached review and the inclusion of more recent literature (e.g. from the year 2022) are suggested.

Author Response

Reviewer 3

Comments and Suggestions for Authors

The article addresses an important topic, is well structured, the data are transformed into important information and the discussion is well grounded.

Some changes indicated in the attached review and the inclusion of more recent literature (e.g. from the year 2022) are suggested.

Thank you for taking the time and effort to review our manuscript. We appreciate your valuable feedback and have carefully considered your comments and suggestions. We have revised the manuscript according to your instructions in the comments and added several references from 2022 and 2023. We hope that our revised manuscript will address all of your concerns and meet the standards of the journal. Thank you again for your time and input.

Round 2

Reviewer 1 Report

This revised manuscript has been improved greatly. All questionable parts have been revised following my prior comments. 

Reviewer 2 Report

I congratulate the authors for the work done.

The manuscript can be accepted to be published in Agronomy Journal.